# Single-Particle Irradiation Effect and Anti-Irradiation Optimization of a JLTFET with Lightly Doped Source

**DOI:** 10.3390/mi14071413

**Published:** 2023-07-13

**Authors:** Haiwu Xie, Hongxia Liu

**Affiliations:** 1Key Laboratory for Wide-Band Gap Semiconductor Materials and Devices of Education, The School of Microelectronics, Xidian University, Xi’an 710071, China; xiehaiwu.love@163.com; 2The School of Physics and Electronic Information Engineering, Qinghai Normal University, Xining 810016, China

**Keywords:** band-to-band tunneling (BTBT), linear energy transfer value (LET), single-particle irradiation effect, anti-irradiation optimization

## Abstract

In this article, the particle irradiation effect of a lightly doped Gaussian source heterostructure junctionless tunnel field-effect transistor (DMG-GDS-HJLTFET) is discussed. In the irradiation phenomenon, heavy ion produces a series of electron-hole pairs along the incident track, and then the generated transient current can overturn the logical state of the device when the number of electron-hole pairs is large enough. In the single-particle effect of DMG-GDS-HJLTFET, the carried energy is usually represented by linear energy transfer value (LET). In simulation, the effects of incident ion energy, incident angle, incident completion time, incident position and drain bias voltage on the single-particle effect of DMG-GDS-HJLTFET are investigated. On this basis, we optimize the auxiliary gate dielectric, tunneling gate length for reliability. Simulation results show HfO_2_ with a large dielectric constant should be selected as the auxiliary gate dielectric in the anti-irradiation design. Larger tunneling gate leads to larger peak transient drain current and smaller tunneling gate means larger pulse width; from the point of anti-irradiation, the tunneling gate length should be selected at about 10 nm.

## 1. Introduction

In nanoscale integrated circuits, leakage current increases exponentially with decreasing device feature size and increasing the circuit integration degree. In recent years, among the low-power devices, TFET has attracted a lot of attention due to its own advantages. TFET has small leakage current and good subthreshold characteristics due to the working mechanism of band-to-band tunneling, which is very suitable for low-power circuit applications. Therefore, it is of great significance to study the radiation reliability performance of TFET devices.

The irradiation effect of TFET is less studied in the published literature. At present, Lili Ding [1,2] of Pardova University in Italy has studied the silicon material TFET, and the irradiation source has been chosen as a 10 keV X-ray. The result shows that the oxide trap charge in the gate dielectric changes when the irradiation effect is affected, which leads to the change in threshold voltage and tunneling voltage of the device. The study also compares the Si-based TFET and FDSOI MOSFET, and the result shows that the TFET device has better anti-irradiation characteristics than the FDSOI device. Avashesh Dubey [3,4] conducts a simulation study on Total Ionizing Dose (TID) of SOI TFET, and the results show that the threshold voltage drift and interface trap charge generated by irradiation environment cannot be ignored, and they have a very important effect on the electrical performance of the device. Therefore, the radiation study of TFET has important guiding significance for the practical application of this kind of device [5,6,7,8,9,10].

With the decreasing feature size of devices, the single-particle irradiation effect gradually becomes the main failure mode of devices in digital applications [11,12,13,14,15,16]. The reason for this is that the power supply voltage and node-point capacitance decrease with the decreasing feature size in integrated circuits, which leads to the reduction in the storage charge of logic circuits. In other words, the node-point capacitance becomes smaller and smaller, so the circuit is more likely to produce logic errors. At the same time, the decrease in feature size will make the parasitic bipolar effect more serious, and the increase in node spacing and integration will enhance the charge sharing effect after heavy ion impact [17,18,19,20].

Generally, the drain bias is set as a fixed value greater than 0 under the OFF-state (Vgs = 0 V) when studying the single-particle transient effect; therefore, the drain terminal voltage is usually larger than other voltages, and there is an inverse PN junction in a P-type TFET at the drain terminal. The strong electric field near the anti-biased junction tends to intensify the single-particle transient effect. When the charged particle is incident, it generates transient current and collected charge at the electrode. If this phenomenon occurs in a storage unit, the collected charge is easy to cause logical upset of the storage unit, that is, single-event upset (SEU) is easy to occur when the collected charge is large enough. Therefore, it is of great practical value to study the transient effects of single particles on the TFET structure [20,21,22,23,24].

In this article, we study the single-particle irradiation effect of the DMG-GDS-HJLTFET structure which has been published in reference [4]. As discussed in reference [4], DMG-GDS-HJLTFET adopts a Gaussian doped source and an InAs/GaAs_0.1_Sb_0.9_ heterostructure, yet an engineered control gate is designed to improve the ON-state current and suppress the OFF-state current simultaneously by dividing the control gate into two parts, namely the tunnel gate (TG) and the auxiliary gate (AG) with work functions of Φ_M1_ and Φ_M2_, where Φ_M1_ > Φ_M2_. Therefore, the conduction band and the valance band of the proposed device at source/channel interface are very close to each other, resulting in a smaller tunneling distance in DMG-GDS-HJLTFET. Moreover, TG with the work function of Φ_M1_ can lower the minimum value of the conduction band at the channel region, which further promotes the tunneling probability in DMG-GDS-HJLTFET. Compared with DMG-HJLTFET using the Ge/Si heterostructure in reference [4], the ON-state current of DMG-GDS-HJLTFET is three orders of magnitude higher than that of DMG-HJLTFET, increasing to up to 4.1 × 10^−4^ A/μm. At the same time, the OFF-state current of DMG-GDS-HJLTFET is only 9.92 × 10^−19^ A/μm, and the subthreshold swing average value of DMG-GDS-HJLTFET is as low as 12.7 mV/Dec. Therefore, DMG-GDS-HJLTFET can be used in future low-power applications.

The cross-section tangent and structure diagram of DMG-GDS-HJLTFET are shown in Figure 1. The proposed structure adopts dual material gate to improve the ON-state and OFF-state current, and a Gaussian doped source is used to simulate the random fluctuation phenomenon of actual impurity doping. The dual material gate can be realized by the method of article [24], and the symmetrical structure can be fabricated by using either molecular beam epitaxy (MBE) or the Metal–organic chemical vapor deposition (MOCVD) method; the method of MBE is especially profoundly researched for building III-V compound semiconductor devices. Based on these analyses, the process flow of the proposed device is as follows: First, the body of the proposed structure can be formed by MBE; Second, HfO_2_ can be deposited at the top of the proposed structure. Then, redundant HfO_2_ can be etched and SiO_2_ needs to be deposited. Afterwards, the gate of the proposed structure can be fabricated. Third, the HfO_2_-SiO_2_ junction and the divided gate can be formed at the bottom of the proposed structure with the same method. At last, the contacts are defined. Figure 2 shows the tentative fabrication flow for DMG-GDS-HJLTFET.

Physical dimensions and electrical properties of DMG-GDS-HJLTFET are discussed in detail in reference [4]. Based on the conclusions in the published literature, we simulate the single-particle irradiation effect of DMG-GDS-HJLTFET in this paper.

All the simulations are carried out using ATLAS Silvaco TCAD version 5.20.2.R. In order to calculate the tunneling current, the nonlocal BTBT model (BBT.NONLOCAL) is activated. A basic analytical formulation for band to band tunneling probability T(E) is shown in Equation (1):(1)TE∝-42m*Eg323eћ(Eg+∆Φ)εsiεoxtoxtsi∆Φ,
where m^∗^ is the effective carrier mass, E_g_ is the bandgap, ΔΦ is the energy range over which tunneling can take place, and t_ox_, t_Si_, ε_ox_, and ε_Si_ are the oxide and silicon film thickness and dielectric constants, respectively. As can be seen from the above formulation, small m^∗^ and small E_g_ are required in source region, and appropriate materials need to be selected in the source/channel interface to ensure the appropriate ΔΦ.

Shockley–Read–Hall related to concentration (CONSRH) is used to account for the minority carrier recombination effects and the presence of highly doped channel. In this regard, the Fermi Statistics (FERMI) model and the band gap narrowing (BGN) model are included. Quantum confinement model given by Hansch (HANSCHQM) is used to consider the increased doping levels and thinner gate oxide in the channel. The Schenk model for trap-assisted tunneling (SCHENK.TUNN) is used to include the important role of trap-assisted tunneling.

Section 2 introduces single-particle irradiation effect of DMG-GDS-HJLTFET. Section 3 shows the anti-irradiation optimization for DMG-GDS-HJLTFET. Section 4 concludes the paper.

## 2. Single-Particle Irradiation Effect of DMG-GDS-HJLTFET

The main components of cosmic radiation are about 83% protons, about 13% alpha particles, about 1% heavy ions, and about 3% galactic cosmic radiation electrons and mesons. The energy of these particles needs to be converted when heavy ion incidence occurs on devices of different sizes. In radiation studies, the ionizing particle is typically described by the linear charge deposition (LCD) value; another common measure of the loss of energy is the linear energy transfer (LET) value. The conversion factor from the LET value to the LCD value is then approximately 0.01 for silicon. So, for instance, an LET value of 25 MeV·cm^2^/mg is equivalent to 0.25 pC/µm. In this paper, we use a different LET value to represent different types of particles, and the LET value can be set for different particles according to the device structure.

In the irradiation phenomenon, heavy ion incidence produces a series of electron-hole pairs along the track, and the transient current generated with these electron-hole pairs is heavily influenced by linear energy transfer value (LET). Next, we discuss the influence of different LETs. It is worth emphasizing that the OFF-state single-particle effect is more serious than that of the ON-state, so the simulation bias is the device OFF-state, i.e., Vgs = 0 V (gate-to-source voltage) and Vds = 0.5 V (drain-to-source voltage). The default condition of single-particle effect simulation for different LETs is that the incident completion time is equal to 2 ps, the incident angle is equal to 90° and the incident position is the auxiliary gate of the device.

Figure 3a shows the transient drain current of DMG-GDS-HJLTFET with a different LET value when Vgs = 0 V and Vds = 0.5 V and the incident position is TG. The peak value of the transient drain current increases with the increase in LET. The peak value of transient drain current is 1.45 × 10^−5^ A/μm when LET = 1 MeV·cm^2^/mg, and the peak value of transient drain current reaches 7.39 × 10^−5^ A/μm when LET = 10 MeV·cm^2^/mg, which is five times higher than that of 1 MeV·cm^2^/mg. At the same time, the pulse width of the transient drain current increases with the LET value. If calculated at 90% of the peak value, the pulse widths of LET = 1 MeV·cm^2^/mg and LET = 10 MeV·cm^2^/mg are 1.58 ps and 2.77 ps, respectively.

Figure 3b shows the transient collected charge of DMG-GDS-HJLTFET with different LET values where the collected charge is obtained by integrating the transient current during the whole simulation time and the transient current occurs in the form of a pulse. The pulse current drops to 0 at 2 × 10^−11^ s, so the value of integral is mostly contributed by the current before the 1 × 10^−11^ s time, and the collected charge stays the same after a specific time. The time for the collected charge to enter the saturation value decreases with the increase in the LET value, and the corresponding time of LET = 10 MeV·cm^2^/mg is only 2.78 ps. Moreover, the saturation value of the collected charge increases with the increase in the LET value; the corresponding value of LET = 10 MeV·cm^2^/mg is 0.55 fC.

Figure 3c,d, respectively, show the change in electric field and potential along the cutline. The electric field increases with the increase in the LET value in the range of 10–22 nm and decreases with the increase in the LET value in the range of 22–30 nm, which is consistent with the current change in Figure 3a. The inset in Figure 3d reflects the change in charge density with the LET value, which is exactly the same as the current change in Figure 3a.

In the single-particle effect, the incident angle affects the generation of transient current and the collected charge, and a study is conducted at the tunneling gate to examine this effect, where 0° means the incidence parallel to the tunneling gate and 90° represents the incidence perpendicular to the tunneling gate; the remaining angles increase from small to large in the counterclockwise direction, as shown in the inset of Figure 4a. Figure 4a shows the transient drain current of DMG-GDS-HJLTFET with different incident angles. It can be observed from Figure 4a that the peak value of the transient drain current is the largest in the case of parallel incidence. The reason for this is that parallel incidence affects not only the tunneling gate but also the auxiliary gate in the simulation setting. The influence of the tunneling gate and the auxiliary gate on the drain current is analyzed, respectively, in reference [4]; thus, the situation in Figure 4a appears. Moreover, both the pulse width and the peak value of transient drain current decrease when the incident angle increases from 30° to 90°, because incident angle = 30° corresponds to the largest influence area in this process. The pulse width and transient drain current peak affect the distribution of the collected charge, as shown in Figure 4b, and the saturation value change in collected charge in Figure 4b is consistent with transient drain current in Figure 4a.

Figure 5a,b, respectively, show the variation of transient drain current and collected charge when the incident position changes from the polar gate to the drain region, where the incident position change is depicted in the inset in Figure 5a. On the whole, the closer the incident position is to the drain electrode, the larger the peak value of the corresponding transient current is, which is in accordance with the characteristics of carrier generation and recombination in this device. Unusually, the peak value of the transient drain current is relatively large when the incident location is hetero-dielectric (between AG and TG). The reason for this is that the AG and TG are simultaneously hit at this position, resulting in a large transient value in the current.

The variation of transient current in Figure 5a determines the change of collected charge in Figure 5b. It can be seen from Figure 5b that the saturation value of the collected charge first increases and then decreases when the incident position is transferred from the polar gate to the drain, and the maximum value of saturation occurs when the incident position is TG.

Figure 6a indicates the transient drain current of DMG-GDS-HJLTFET with different incident completion times where the incident position is TG. Here, the incident completion time goes from 0 ps to 10 ps with the step size of 2 ps. It can be seen that the peak value of the transient drain current increases before tp = 6 ps and obviously attenuates after tp = 8 ps; then, it returns to the level before incidence when tp = 10 ps (tp = 0 ps represents no single-particle irradiation). This change is reflected by a potential variation in Figure 6b. The scope of 12 nm–30 nm is the region of tunneling and tunneling gate in this device, and more non-equilibrium carriers can be generated under greater potential. In addition, long incident completion time means the generated non-equilibrium carriers are more likely to be recombined when they pass through the channel, contributing to the transient drain current changes in Figure 6a.

Figure 7a shows the influence of drain voltage on transient current, where the LET value is fixed at 10 MeV·cm^2^/mg and the incident position is selected as TG. The peak value of transient drain current increases with the increase in Vds, which varies from 5.65 × 10^−5^ A/μm to 1.23 × 10^−4^ A/μm when Vds increases from 0.2 V to 0.8 V; however, the pulse width of the transient drain current decreases with the increase in Vds, which decreases from 2.89 ps to 0.4 ps. The reason for this phenomenon is that high drain voltage means high channel potential. More non-equilibrium carriers can be produced in the channel when the drain voltage is higher. At the same time, the generated non-equilibrium carriers are more likely to be collected by the drain voltage at higher values, which will affect the amount of charge collected by the gate. Figure 7b shows the influence of Vds on the collected charge. It can be seen that the non-equilibrium carrier generation process caused by the channel potential is dominant when Vds ≤ 0.6 V, while the drain electrode collection process is superior when Vds ≥ 0.6 V.

## 3. Anti-Irradiation Optimization for DMG-GDS-HJLTFET

In the previous part, the influence of incident angle, incident position, incident completion time and drain bias voltage on the single-particle irradiation effect of DMG-GDS-HJLTFET is analyzed in detail. The results show that the impact of drain region incidence and auxiliary gate incidence is the most obvious, while the doping concentration in the drain region cannot be changed; therefore, we adjust the type of the auxiliary gate dielectric and the length of the auxiliary gate for DMG-GDS-HJLTFET anti-irradiation optimization.

Figure 8a shows the transient drain current of DMG-GDS-HJLTFET with different auxiliary gate dielectric. It is clear that the dielectric type under the auxiliary gate has an obvious influence on the peak value and pulse width of the transient drain current. The peak value decreases and the pulse width increases with the increase in the dielectric constant. The details are shown in Table 1.

It can be observed from Table 1 that a larger auxiliary gate dielectric constant helps to prevent the transient current generated by the single-particle irradiation effect. However, a larger auxiliary gate dielectric constant means a wider pulse width, so it is necessary to compare the saturation value of the collected charge. In Figure 8a, the peak value of the transient drain current decreases with the increase in the dielectric constant while the pulse duration increases with the increase in the dielectric constant. Based on these two aspects, the change in collected charge is shown in Figure 8b. In Figure 8b, there is a maximum collected charge saturation value when the auxiliary gate dielectric is Al_2_O_3_. The reason for this is that a larger peak current and a wider pulse width occurs when the dielectric layer is chosen as Al_2_O_3_. However, the pulse is wider when the auxiliary gate dielectric is HfO_2_, which can effectively suppress the peak current, resulting in a smaller saturation value of the collected charge. Therefore, in order to effectively prevent the single = -particle irradiation effect, HfO_2_ with a larger dielectric constant should be selected as the auxiliary gate dielectric in the anti-irradiation design.

Figure 9a indicates the transient drain current generated by single-particle irradiation in DMG-GDS-HJLTFET when the tunneling gate length increases with a step size of 2 nm within the range of 3 nm to 17 nm (correspondingly, the auxiliary gate length changes from 17 nm to 3 nm). It can be observed from Figure 9a that a longer tunneling gate can produce a greater peak value of the transient drain current. The peak value of the transient drain current is 3.52 × 10^−5^ A/μm when TG = 3 nm while the peak value of the transient drain current reaches 1.11 × 10^−4^ A/μm when TG = 17 nm, which increases by more than three times. However, the change in pulse width is just the opposite, and its value gradually decreases with increasing the length of the tunneling gate. Calculated by 90% of the pulse peak value, the pulse width is 2.82 ps when TG = 3 nm and the pulse width is 1.24 ps when TG = 17 nm.

Figure 9b shows the collected charge under different tunneling gate lengths. The change in collected charge is related to both the peak value and the pulse width of the transient drain current, that is, the collected charge is the integral of the transient current with respect to time, and its magnitude is positively related to the area formed by the transient drain current and the time axis. The current variation in Figure 9a determines the collected charge distribution in Figure 9b. The saturation value of the collected charge is the largest when TG = 5 nm, which is consistent with the current change in Figure 9a. In the case of single-particle irradiation, a larger tunneling gate leads to a larger peak value of the transient drain current while a smaller tunneling gate means a larger pulse width. Therefore, the appropriate length of the tunneling gate is about 10 nm.

## 4. Conclusions

In this paper, the single-particle irradiation effect of DMG-GDS-HJLTFET is studied. We discuss the performance under different LETs, different incident angles, different incident completion times, different incident positions and different drain bias voltages in detail. Results show that the peak value of the transient drain current of DMG-GDS-HJLTFET is 7.39 × 10^−5^ A/μm when LET = 10 MeV·cm^2^/mg. If calculated at 90% of the peak value, the pulse widths of LET = 1 MeV·cm^2^/mg and LET = 10 MeV·cm^2^/mg are 1.58 ps and 2.77 ps, respectively. The pulse width and the peak value of the transient drain current decrease with the increase in the incident angle within 30°–90°, and a closer incident position to the drain electrode can generate a larger peak value of the corresponding transient current. Moreover, the peak value of the transient drain current returns to the level before incidence when tp = 10 ps, and the peak value of the transient drain current increases with the increase in Vds. Based on the comparison of the single-particle irradiation characteristics of DMG-GDS-HJLTFET, the anti-irradiation analysis and optimization are carried out. Results show that HfO_2_ with a larger dielectric constant should be selected as the auxiliary gate dielectric in the anti-irradiation design in order to effectively prevent the single-particle irradiation effect, and the appropriate length of the tunneling gate is about 10 nm.

## Figures and Tables

**Figure 1 micromachines-14-01413-f001:**
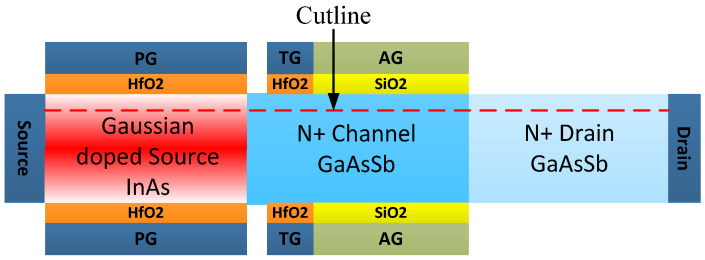
Cross-section tangent and structure diagram of DMG−GDS−HJLTFE.

**Figure 2 micromachines-14-01413-f002:**
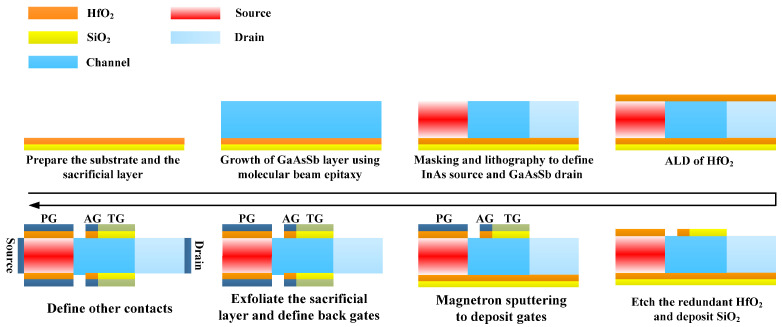
Tentative fabrication flow of DMG−GDS−HJLTFET.

**Figure 3 micromachines-14-01413-f003:**
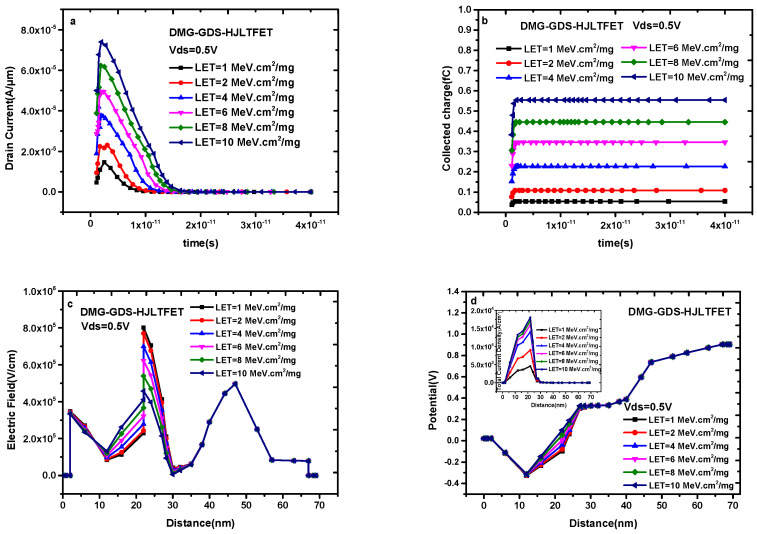
(**a**) Transient drain current of DMG−GDS−HJLTFET; (**b**) transient collected charge of DMG−GDS−HJLTFET; (**c**) change of electric field along the cutline and (**d**) change in potential along the cutline with different LET.

**Figure 4 micromachines-14-01413-f004:**
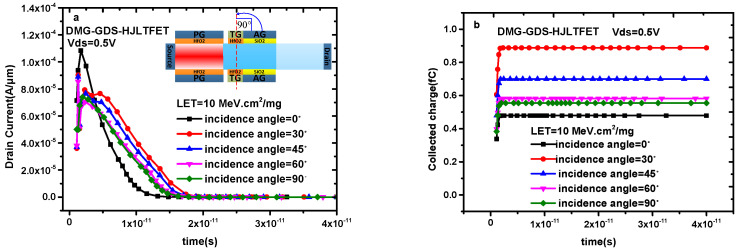
(**a**) Transient drain current of DMG−GDS−HJLTFET and (**b**) transient collected charge of DMG−GDS−HJLTFET with different incident angles (0°, 30°, 45°, 60°, 90°).

**Figure 5 micromachines-14-01413-f005:**
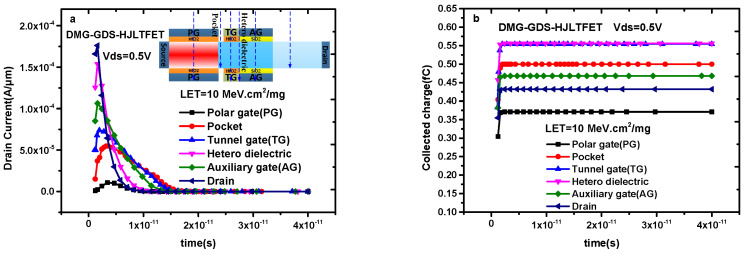
(**a**) Transient drain current of DMG−GDS−HJLTFET and (**b**) transient collected charge of DMG−GDS−HJLTFET with different incident position.

**Figure 6 micromachines-14-01413-f006:**
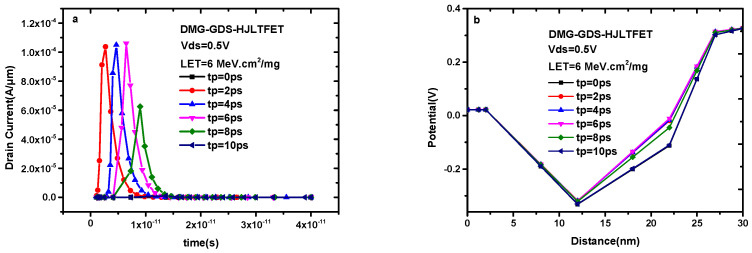
(**a**) Transient drain current of DMG−GDS−HJLTFET and (**b**) corresponding curve of electric potential change with different incident completion time.

**Figure 7 micromachines-14-01413-f007:**
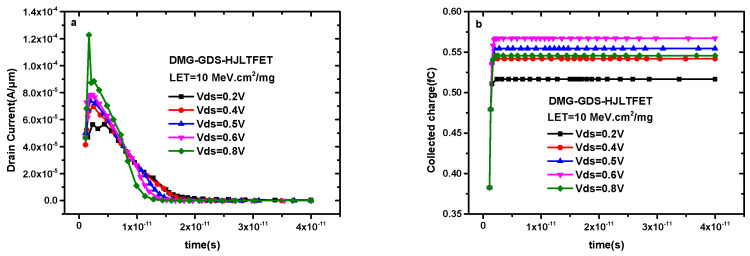
(**a**) Transient drain current of DMG−GDS−HJLTFET and (**b**) transient collected charge of DMG−GDS−HJLTFET with different drain voltage.

**Figure 8 micromachines-14-01413-f008:**
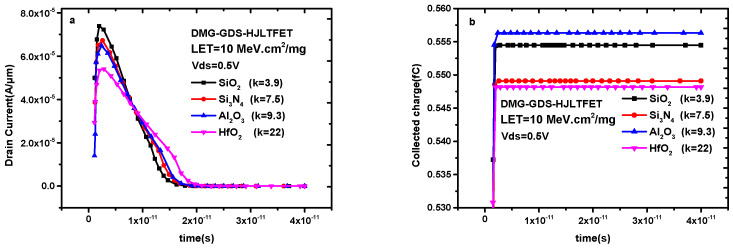
(**a**) Transient drain current of DMG−GDS−HJLTFET and (**b**) transient collected charge of DMG−GDS−HJLTFET with different auxiliary gate dielectric.

**Figure 9 micromachines-14-01413-f009:**
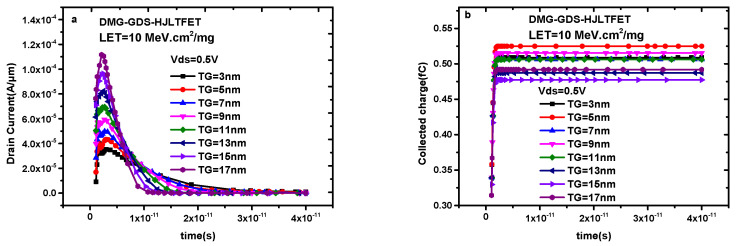
(**a**) Transient drain current of DMG−GDS−HJLTFET and (**b**) transient collected charge of DMG−GDS−HJLTFET with different auxiliary gate length.

**Table 1 micromachines-14-01413-t001:** Comparison of transient drain current and collected charge of DMG-GDS-HJLTFET with different auxiliary gate dielectric.

Dielectric Type	SiO_2_	Si_3_N_4_	Al_2_O_3_	HfO_2_
Dielectric constant	3.9	7.5	9.3	22
Peak value (A/μm)	7.39 × 10^−5^	6.73 × 10^−5^	6.48 × 10^−5^	5.41 × 10^−5^
Pulse duration (s)	2.59 × 10^−11^	2.64 × 10^−11^	2.67 × 10^−11^	2.91 × 10^−11^
Collected charge (fC)	0.55	0.549	0.556	0.548

## Data Availability

The simulation data and the method will be shared upon request to the authors.

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
