# Peer review of "Single-Particle Irradiation Effect and Anti-Irradiation Optimization of a JLTFET with Lightly Doped Source"

_micromachines, 2023, doi:10.3390/mi14071413_

Round 1

Reviewer 1 Report

The authors investigated the the single particle irradiation effect of DMG-GDS-HJLTFET, including the impact of different LETs,  incident angles, incident completion times, incident positions and different drain bias voltages.  Analysis and design optimization for irradiation hardening are carried out. There are several suggestions to further improve the manuscript.  

1) As an independent piece of work, there should a short paragraph to briefly introduce the device, including the operating principles, applications, performance against benchmark devices, etc. 

2) There should also a brief introduction of the simulation models, physics and equations used in the software. 

3) It  may also be worthwhile to discuss the impact of different types of particles.

4) Please discuss whether it is realistic to achieve 10 nm in this rather complex structure?

5) English can be improved. 

English can be improved with some editing.

Reviewer 2 Report

Fabrication steps of the proposed device should be included  in the manuscript with proper diagrams.

Fabrication steps of the proposed device should be included  in the manuscript with proper diagrams.

Reviewer 3 Report

The article represents the original work of the authors in the area of single particle irradiation effect and anti-irradiation optimization of a junctionless tunnel field-effect transistor. In the paper is simulated the single particle irradiation effect of a lightly-doped Gaussian source hetero structure junctionless tunnel field-effect transistor.

The topic and presented data are interesting and the investigation is attractive. The presented research is important for practical application and obtained results can contribute to a further analysis of investigated effects.

Considering that interesting research is presented the paper should be accepted (after minor corrections).

Some suggestions to authors:

All Figures (except Figure 1) should be somewhat larger. This will allow the figures to be more visible and clearer. Especially in the printed version, it is now not possible to clearly distinguish the data. It is particularly difficult to see the data on the inserted figures

Only 4 references are mentioned in the text of the paper. However, there are 23 of them in the list of references. Therefore, other references should be referred to in the paper.

Figure 2 should be mentioned in the text before the displayed Figure 2. The same goes for the Figures 6-8.

1 MeV.cm2/mg    it should be   1 MeV.cm2/mg.

In general (for all data in the paper), it is common to have a small space between the number and the unit. Also, it is not usual that word “Figure” is in one line and number is in in another, like in lines 146, 147, 148, 219.  The same goes for the numbers - for example in lines 173 and 174.   

The English language is mostly good, but minor editing of English language required.

Reviewer 4 Report

This article investigates the single particle irradiation effect and anti-irradiation optimization for JLTFET. The following issues should be addressed:

1. Is it possible to compare with simulation results based on actual experiment results? 

2. “drain terminal voltage is usually larger than other voltage” in line 55 on Page 2 should be further explained. The effect of the single particle effect on the drain terminal compared to other terminals and the additional explanation of focusing on the single particle rather than the multiple particles are required. 

3. The insert figures of Figure 2. (d), Figure 3. (a), Figure 4. (a) needs improvement for the reader. 

4. In most figures, further clarification is required if there is a reason why the collected charge stays the same value after a specific time, i.e., 1x10^-11.

The quality of the English language is good.

Round 2

Reviewer 2 Report

The authors have addressed all the reviewer comments. This can be accepted in the current form.